# DataSEA - An Automatic Framework for Comprehensive Dataset Processing Using Large Language Models

## Abstract

In the era of data-driven decision-making, efficiently acquiring and analyzing diverse datasets is critical for accelerating research and innovation. Yet, traditional manual approaches to dataset discovery, preparation, and exploration remain inefficient and cumbersome, especially as the scale and complexity of datasets continue to expand. These challenges create major roadblocks, slowing down the pace of progress and reducing the capacity for data-driven breakthroughs. To address these challenges, we introduce DataSEA (Search, Evaluate, Analyze), a fully automated system for comprehensive dataset processing, leveraging large language models (LLMs) to streamline the data handling pipeline. DataSEA autonomously searches for dataset sources, retrieves and organizes evaluation metadata, and generates custom scripts to load and analyze data based on user input. Users can provide just a dataset name, and DataSEA will handle the entire preparation process. While fully automated, minimal user interaction can further enhance system accuracy and dataset handling specificity. We evaluated DataSEA on datasets from distinct fields, demonstrating its robustness and efficiency in reducing the time and effort required for data preparation and exploration. By automating these foundational tasks, DataSEA empowers researchers to allocate more time to in-depth analysis and hypothesis generation, ultimately accelerating the pace of innovation. The code is available at https://github.com/SingleView11/DataSEA.

## 1 Introduction

In the age of artificial intelligence, the significance of data is undeniable. The volume of data generated and published online is rapidly increasing, yet searching for structured data on the internet remains challenging [Kacprzak et al. (2018)]. On one hand, quickly and clearly understanding the structure and content of large datasets—spanning social sciences, life sciences, high-energy physics, climate science, and other fields—has become increasingly difficult. On the other hand, as the scale and complexity of data grow, the efficiency of manually searching for and downloading datasets diminishes, turning into a daunting task. Traditional manual methods for discovering, preparing, and exploring data are becoming increasingly cumbersome and inefficient. Consequently, swiftly acquiring and analyzing diverse datasets has become a crucial factor in driving research and innovation.

With the advancement of prompt engineering, large language models (LLMs) have demonstrated impressive performance across various fields [Wei et al. (2022); Kojima et al. (2022); Wang et al. (2022); Zhou et al. (2022); Madaan et al. (2024); Bai et al. (2022); Chen et al. (2023)]. The strong capability of LLMs to process vast amounts of textual information opens new possibilities for the automated discovery, evaluation, and analysis of datasets. By utilizing LLMs to analyze the web information related to specific datasets, it is possible to organize this data into structured formats, facilitating further computational processing and user interaction.

Thus, we propose DataSEA (Search, Evaluate, Analyze), a comprehensive automated dataset processing system based on large language models. Additionally, by treating the collected data as a

new corpus for LLMs, it can enhance user interaction, further improving the system's accuracy and adaptability to specific data processing needs. This allows users to gradually understand the various characteristics of target datasets and perform diverse evaluations in a question-answering system-like environment.

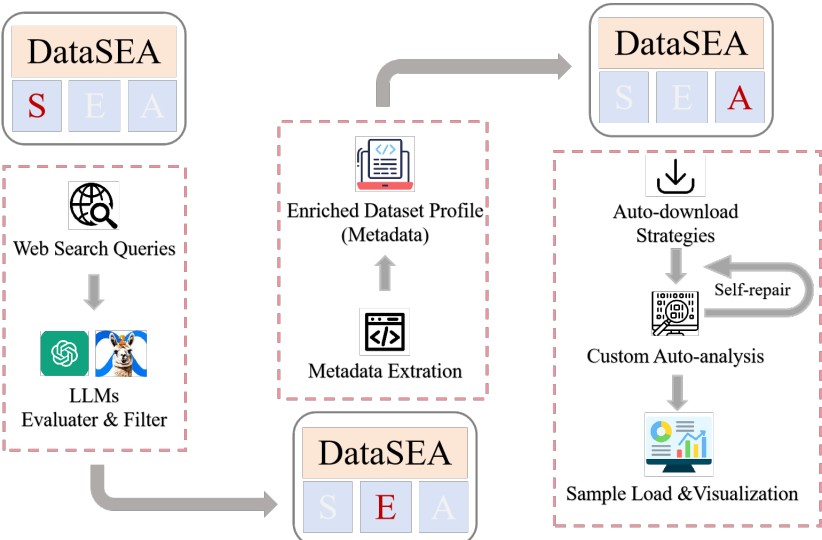

Figure 1: The architecture of DataSEA. The system is divided into three key modules: Search, Evaluate, and Analyze. The Search Module (S) retrieves top results from search engines and evaluates the relevance of the links using LLM models. The Evaluate Module (E) extracts metadata from the identified websites and retrieves research papers citing the dataset, followed by metadata optimization. The Analyze Module (A) generates and executes code to download datasets, hypothesizes possible download methods, and visualizes data samples. The entire process can be fully automated, though users may intervene to improve accuracy and filter unwanted downloads.

The design goal of DataSEA is to simplify and accelerate the data processing workflow, including the search, evaluation, and analysis of datasets. Users only need to provide a dataset name, and DataSEA can autonomously perform the search for dataset sources, retrieve and organize metadata, and generate custom scripts to load and analyze data based on user requirements.

To demonstrate the significant practical value of this system, we evaluated DataSEA on multiple datasets across various fields, such as computer vision, natural language processing, speech recognition, medicine, natural sciences, social sciences, and finance. Accessing these datasets is crucial for promoting the reproducibility of research findings, enabling scientists to build upon the work of others, and facilitating easier access to information and its sources for data journalists [Brickley et al. (2019)]. Experimental results indicate that DataSEA exhibits strong robustness and efficiency, significantly reducing the time and effort required for data preparation and exploration. Additionally, DataSEA provides rich and logical visualization, evaluation, and interpretation of data, greatly lowering the cost of understanding the content of datasets.

We also tested the performance of various LLMs as core processing components of DataSEA, including GPT-4 [Achiam et al. (2023)], GPT-3.5 [Ye et al. (2023)], and Llama [Touvron et al. (2023)]. Our findings indicate that the effectiveness of different LLM models in dataset collection, evaluation, and analysis is related to their parameter counts, which influences their ability to integrate and comprehend web information. This suggests that as model parameters increase in the future, alongside improvements in long text processing and logical reasoning capabilities, DataSEA is likely to exhibit enhanced capabilities in dataset collection and processing.

In Section 2, we outline related work; Section 3 elaborates on the methods used to construct DataSEA; Section 4 presents experiments demonstrating the effectiveness of DataSEA; Section 5 discusses the existing limitations; and finally, Section 6 provides a comprehensive summary of this paper. Additional relevant data can be found in the appendix.

## 2 RELATED WORK

**Efficient Dataset Discovery and Exploration**  Efficient dataset discovery, preparation, and exploration are critical components of the data-driven research pipeline. However, traditional approaches often require significant manual effort, involving labor-intensive tasks such as searching for relevant datasets, preparing them for analysis, and creating appropriate exploratory tools. Recent advances in automated data management have sought to alleviate these challenges by streamlining data discovery, cleaning, transformation, and exploration. Automated data integration tools such as advanced dataset search systems Brickley et al. (2019) and AutoML frameworks Zöller & Huber (2021) have demonstrated considerable promise in minimizing manual workload, though many solutions still face limitations regarding the flexibility and comprehensiveness of the automated workflow. These advancements reflect a growing interest in reducing human intervention and enhancing efficiency through intelligent data management solutions.

**Automated Dataset Processing Frameworks**  Several automated dataset processing tools have emerged to address various parts of the data handling workflow. Tools such as Trifacta and OpenRefine Petrova-Antonova & Tancheva (2020) focus on data wrangling, emphasizing interactive user experiences for cleaning and transforming data. Although these tools significantly improve the efficiency of data preprocessing, they require extensive user involvement throughout the process and lack fully automated workflows, particularly in terms of dataset discovery and evaluation.

The development of systems like AutoML He et al. (2021) has further paved the way for automation by addressing tasks like feature engineering and model selection. However, while AutoML tools effectively handle model training and hyperparameter tuning, they often depend on structured, pre-prepared datasets. The processes of discovering datasets and assessing their suitability for analysis largely remain manual, limiting the overall automation potential in the data science pipeline Biswas et al. (2022).

**Leveraging Language Models for Automation**  LLM models such as GPT-4 Achiam et al. (2023) and LLAMA Touvron et al. (2023), have demonstrated significant capabilities in understanding natural language, generating code, and automating workflows in complex domains. Previous research has leveraged LLMs to generate scripts for data processing Biswas & Talukdar (2024); Nejjar et al. (2023); Patiny & Godin (2023), streamlining the creation of custom data handling scripts. These efforts highlight the potential of LLMs in automating repetitive tasks, but they often focus narrowly on code generation without addressing the end-to-end dataset processing pipeline.

The recent work on LLM-based assistants (e.g., GPT-4, LLAMA) has further demonstrated the applicability of these models for responding to natural language queries related to data analytics Ram et al. (2024), offering on-demand support for exploratory data analysis (EDA) Ma et al. (2023) and visualization Sah et al. (2024). However, these applications are reactive, requiring substantial user intervention in specifying datasets, parameters, and contexts for each step.

**Comprehensive Dataset Automation**  DataSEA builds on these advancements, aiming to deliver a fully automated framework for dataset processing that encompasses not only code generation but also dataset discovery and evaluation. Unlike existing semi-automated tools that require significant human interaction, DataSEA autonomously manages dataset discovery, metadata extraction, and script generation, reducing the need for user input to a minimum. Data-centric AI Zha et al. (2023) suggests that focusing on automating data-handling processes can significantly accelerate research outcomes Mittal et al. (2023), and DataSEA aligns closely with this vision by implementing an automated pipeline that integrates search, evaluation, and analysis.

By leveraging LLMs to automate not only the preparation but also the discovery and evaluation of datasets, DataSEA contrasts with existing solutions that focus predominantly on either preparation or analytics. This comprehensive approach empowers researchers to reduce the time spent on foundational tasks, allowing for more in-depth analysis and exploration of the data.

**Summary**  While previous work has made significant strides in automating parts of the data preparation and analysis workflow, DataSEA is among the first to provide a fully integrated solution for dataset discovery, evaluation, and custom analysis using large language models. By autonomously handling these key stages, DataSEA extends beyond the capabilities of current LLM-driven tools and represents a significant step toward automating the entire data lifecycle. This approach aligns

with recent trends in AI-driven automation and data-centric methodologies, ultimately accelerating the pace of innovation in data-driven research.

# 3 METHODOLOGY

## 3.1 SYSTEM OVERVIEW

DataSEA is composed of three core modules: Search, Evaluate, and Analyze. The system leverages large language models to intelligently locate dataset sources, extract metadata, and generate custom scripts for loading and visualizing the data. Users can input a dataset name and optional description, and DataSEA autonomously handles the remainder of the process. The architecture allows for minimal user interaction, but additional input can improve the system's accuracy.

To enhance the effectiveness of LLMs, DataSEA employs instruction-prompting [Brown (2020)] and a multi-chunk strategy [Liu et al. (2024)] to handle long inputs, ensuring that even large datasets can be processed effectively by breaking the data into manageable sections while maintaining context across chunks.

## 3.2 SEARCH MODULE

The Search Module in DataSEA automates the process of discovering dataset websites by leveraging search engines and LLMs to filter and rank relevant results. The user starts by inputting a dataset name and, optionally, additional dataset details to refine the search. Drawing inspiration from tools like Google Dataset Search [Brickley et al. (2019)], the system generates optimized search queries based on the input and send them to search engines such as Google to retrieve the top-ranking links.

Once the top links are retrieved, the system performs web content extraction on each page. The contents are then analyzed by the LLM, which generates evaluation info. Similar to work on using LLMs for content understanding and retrieval tasks [Brown (2020)], the LLM helps filter out irrelevant or low-quality pages. The links are ranked based on their evaluation info, with the top-ranking results being those most likely to contain useful dataset information. More detail can be found in Appendix.

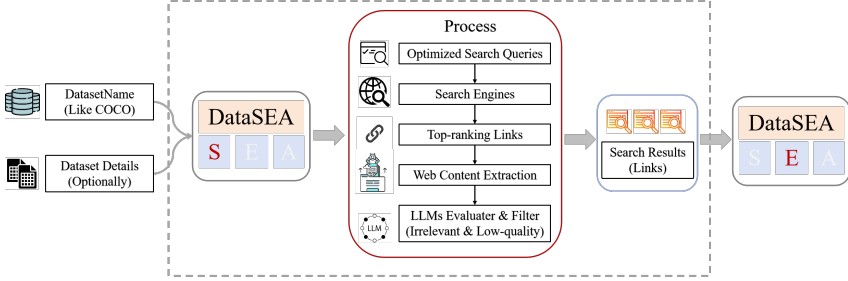

Figure 2: The process flow of the Search Module in DataSEA. The system retrieves top links from search engines based on user input, evaluates the relevance of each link using the LLM model, and filters out irrelevant or low-quality pages. The links are ranked by relevance to the dataset, allowing the user to quickly access accurate and useful resources.

## 3.3 EVALUATION MODULE

The Evaluation Module in DataSEA generates the metadata of the dataset, including various information with 3 steps: Metadata Extraction, Reference Paper Retrieval, Metadata Extension. User can customize the properties of the metadata template.

### 3.3.1 METADATA EXTRACTION

In the first step, the system processes the links identified as relevant in the Search Module. The system extracts their website content and uses the LLM to get metadata. This process is guided by a

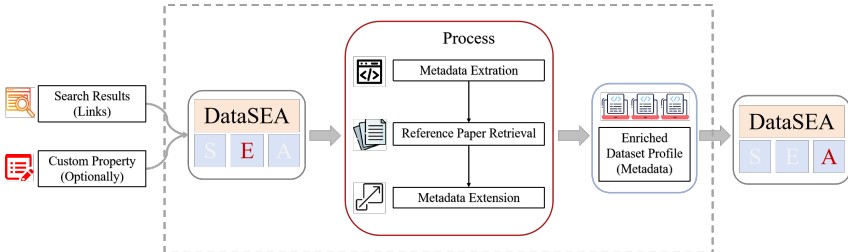

Figure 3: The process flow of the Evaluation Module in DataSEA. The system extracts metadata from relevant links, retrieves research papers that reference the dataset, and validates the relevance of each paper using LLM models. The metadata is then optimized and extended using information from the papers, ensuring a comprehensive dataset profile.

preset of metadata attributes, including the dataset's usage, content and scale, application fields, and other important factors. Additionally, users have the flexibility to input custom property names, and the system dynamically optimizes prompts for the LLM to retrieve those specific properties. The results from different links are combined at the end for optimization.

### 3.3.2 REFERENCE PAPER RETRIEVAL

The second stage focuses on retrieving and ranking research papers that reference the dataset. The system uses the Google Scholar API to find papers that may have cited the dataset. These papers are ranked by citation count, following the widely accepted practice of using citation metrics as indicators of a paper's impact (Bornmann & Daniel, 2008). For each paper, the metadata extracted in the first step is used to validate whether the paper indeed references the correct dataset, as there may be cases of duplicate names or other inaccuracies. The system filters the most impactful papers, and the user can specify the number of papers to be collected.

### 3.3.3 METADATA EXTENSION

In the final step, the system extracts additional metadata from the validated reference papers. This may include more detailed descriptions of the dataset's features, specific application examples, and additional context provided by the authors. The extracted metadata is then combined with the original information from the dataset's website, resulting in an enriched and comprehensive dataset profile.

### 3.4 ANALYSIS MODULE

The Analysis Module in DataSEA is used to download the dataset and analyze it by generating code and test them. The final generated code can load and visualize dataset samples, and user can input a customized requirement and the code generation will be adapted to satisfy the requirement.

### 3.4.1 DATASET DOWNLOAD

In this phase, the module utilizes metadata from the Evaluation Module to generate hypotheses for downloading the dataset from the identified websites. For each website containing dataset information, the system generates hypotheses regarding possible download methods. It then creates code by combining the hypothesis, the website content, and the dataset metadata, and executes this code to attempt to download the dataset.

The generation of hypotheses has proven to be an effective method for handling uncertainty in data retrieval processes, as it allows the system to explore multiple potential download strategies simultaneously. This approach is inspired by the inductive reasoning capabilities of language models, as demonstrated in Hypothesis Search (Wang et al., 2023), where generating and testing multiple hypotheses leads to more robust and successful outcomes.

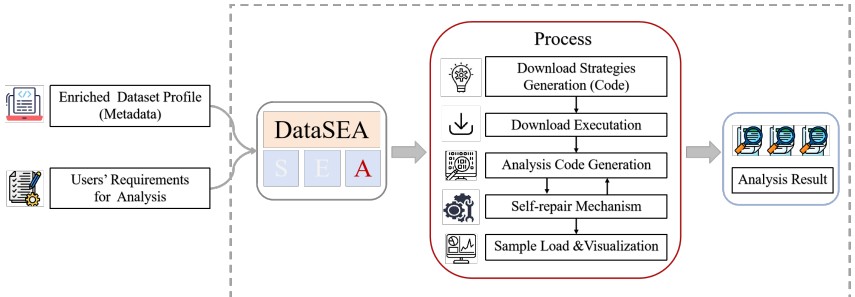

Figure 4: The process flow of the Analyze Module in DataSEA. The system generates hypotheses for dataset download based on metadata, executes the download, and analyzes the dataset by reading the samples of the raw data. It then generates visualization code, with options for manual intervention to refine the analysis. The system also incorporates a self-repair feature to handle any issues in code execution.

Since multiple links and hypotheses may be involved, the system organizes the download results into separate folders, each corresponding to a specific hypothesis. Users can manually inspect the results and delete unwanted downloads, such as cases where multiple datasets (e.g., raw, processed) are available and only a portion is needed. This flexibility allows the user to refine the dataset collection process and accelerate further analysis.

### 3.4.2 DATASET ANALYSIS AND VISUALIZATION

After successfully downloading the datasets, the system generates custom analysis code by calling the LLM with prompts that include the dataset metadata and initial portions of the downloaded data. This analysis code is designed to load the dataset, read the samples of the raw dataset, and visualize key aspects of the dataset's structure and contents.

The analysis code generation process incorporates a self-repair mechanism, drawing from approaches like CodeT5 (Wang et al., 2021). If the generated code fails during execution, the system automatically collects the error log and combines it with the original code to form a more detailed context. This context is sent back to the LLM, which attempts to identify and fix the issues in the code. The system iterates through this feedback loop until a working version of the code is produced, significantly improving the reliability and robustness of the analysis code. This feature allows the system to autonomously handle failures and continuously improve the generated code without requiring user intervention.

In addition to the automated processes, users have the ability to write customized requirements. The system will generate and test code based on the user's input, allowing for tailored analysis that fits specific research needs. This user interaction complements the fully automated pipeline, providing flexibility for users to guide the analysis toward more specific goals if needed.

## 4 EXPERIMENTS

### 4.1 SETUP

**Datasets** We evaluated DataSEA on 100 datasets across various fields to assess its generalizability and effectiveness in automating dataset discovery, evaluation, and analysis. The datasets were sourced from repositories such as Google Dataset Search, Kaggle, and other publicly available platforms. These datasets span a wide range of fields, including Computer Vision, Natural Language Processing (NLP), Healthcare, Speech and Audio, Natural Sciences, Social Science, Finance, Transportation, Recommendation Systems, Time Series Analysis, Robotics, and Agriculture. The diversity in size, format, and complexity of these datasets allowed for a comprehensive evaluation of DataSEA's performance across different domains.

**Models** We used three models to evaluate the performance of DataSEA: gpt-4o, gpt-4o-mini, and llama3. For gpt-4o and gpt-4o-mini, we directly call openai apis; for llama3, we deploy it locally.

**Parameters** We configured three different modes in DataSEA to explore the trade-offs between speed and accuracy: High-Speed, Medium-Speed and Slow-Speed version.High-Speed Version is optimized for fast dataset discovery and analysis by reducing the number inhyper parameters. Medium-Speed Version is a balance between performance and speed. Slow-Speed Version is focused on thorough dataset discovery and analysis. The 3 modes take about 3-5 / 10-15 / 20-60 minutes.

The hyper parameters include the number of websites collected per dataset, the number of hypotheses generated, the number of relevant papers retrieved, the number of download code generation trial, the number of analysis code idea and the number of self-repairs performed when issues were encountered in code generation.

**Evaluation Metrics** We evaluated DataSEA based on the performance of each of its three core modules: Search (S), Evaluate (E), and Analyze (A). For every module we have different metrics, with more detail in the next subsection.

## 4.2 MAIN RESULTS

We present the evaluation of each module—Search (S), Evaluate (E), and Analyze (A)—using three model and their high, medium, and slow-speed versions. Each module's performance is detailed in its respective section.

**Search Module** The Search Module was evaluated based on its ability to find relevant websites. The main evaluation metrics are the **RWF** (Relevant Websites Found in LLM return, true if one is found otherwise false), **ACC** (Relevance judgement accuracy of the LLM).

| Model | Version | RWF (%) | ACC (%) |
|---|---|---|---|
| gpt-4o | High Speed | 100 | 92.36 |
| | Medium Speed | 100 | 93.02 |
| | Slow Speed | 100 | **96.33** |
| gpt-4o-mini | High Speed | 97 | 82.77 |
| | Medium Speed | 98 | 84.14 |
| | Slow Speed | 98 | 90.13 |
| llama3 | High Speed | 99 | 83.20 |
| | Medium Speed | 100 | 81.82 |
| | Slow Speed | 100 | 87.67 |

Table 1: Results for Search Module (S) across different models and versions.

The results show that gpt-4o under low-speed mode achieves the highest accuracy for relevant websites judging. As for the false judgements, most cases are false negative - the website does contain information about the dataset but it is judged as not because there are too many redundant information in the html content, so that the information about the dataset is stuck in the middle and not captured well by the llm. This result is just like the phenomenon described in a paper showing LLM's incapability in processing long context[Liu et al. (2024)].

**Evaluation Module** The Evaluation Module was assessed using the quality of generated metadata and retrieved papers. The **I-ACC** (Initial Metadata Accuracy) reflects the system's ability to extract correct metadata across different properties. It is calculated by averaging the accuracy of the 8 different properties in the metadata. We also evaluated **R-ACC** (Relevant Papers Accuracy, if a judged reference paper is really referring to the dataset) and the **E-ACC** (Extended Metadata Accuracy).

**Analysis Module** For the Analysis Module, we focused on the **DDS** (Dataset Download Success), **H-ACC** (Hypothesis Accuracy), **CAS** (Code Analysis Success Without Intervention), and **CAS-I** (Code Analysis Success With Intervention). The H-ACC is examined by manually following the steps generated by a hypothesis, and is marked as true if there exist one true hypothesis.

The intervention in code analysis means downloading the dataset by following the hypothesis, because while the system can be fully-automated, it is hard to download datasets automatically as most

| Model | Version | I-ACC (%) | R-ACC (%) | E-ACC (%) |
|-------|---------|-----------|-----------|-----------|
| gpt-4o | High Speed | 92.63 | **87.31** | 98.00 |
| | Medium Speed | 93.13 | 84.09 | 98.00 |
| | Slow Speed | **94.75** | 85.12 | **98.25** |
| gpt-4o-mini | High Speed | 87.13 | 80.94 | 93.50 |
| | Medium Speed | 85.00 | 82.30 | 92.25 |
| | Slow Speed | 90.38 | 78.64 | 93.50 |
| llama3 | High Speed | 82.63 | 81.53 | 84.75 |
| | Medium Speed | 83.00 | 81.20 | 84.13 |
| | Slow Speed | 83.00 | 84.22 | 85.88 |

Table 2: Results for Evaluation Module (E) across different models and versions.

datasets' host platform will require login or email request for the dataset, and can not be crawled easily. Even if the datasets are available publicly, there may be multiple datasets with different properties(raw, processed, etc), and serve different purposes, so a comparison for code analysis between automatically downloaded data and manually downloaded data following the hypothesis is needed.

| Model | Version | DDS (%) | H-ACC (%) | CAS (%) | CAS-I (%) |
|-------|---------|---------|-----------|---------|-----------|
| gpt-4o | High Speed | 9 | 81 | 11 | 32 |
| | Medium Speed | 11 | 88 | 12 | 35 |
| | Slow Speed | **12** | **91** | **15** | **38** |
| gpt-4o-mini | High Speed | 4 | 76 | 9 | 28 |
| | Medium Speed | 6 | 82 | 9 | 32 |
| | Slow Speed | 6 | 87 | 9 | 31 |
| llama3 | High Speed | 2 | 62 | 3 | 19 |
| | Medium Speed | 3 | 69 | 3 | 21 |
| | Slow Speed | 4 | 70 | 3 | 21 |

Table 3: Results for Analysis Module (A) across different models and versions.

The interesting finding is that sometimes the CAS is higher than H-ACC, which is counter-intuition because it is hard to imagine analyzing a dataset when it is not downloaded. This is due to the prompt design, as instruction to try to generate code for the dataset without checking the dataset info. As a result, for some popular datasets like MNIST or CIFAR-10, even though the download dataset folder is empty, the generated code can still be run and will successfully generate visualization of samples.

The CAS-I will be lifted greatly if user manually follow the ideas generated and download the dataset. For example, downloading a dataset in Kaggle is convenient and only need a click of button if user is logged in, but the system currently cannot auto-login for the user and will fail to download the dataset.

## 5 LIMITATIONS

DataSEA faces several challenges, including its inability to process databases and databanks, which limits its application in biological fields like genomics [Sherry et al. (2001) ] and proteomics [Abola et al. (1984)]. Additionally, its performance depends heavily on LLMs, and the system exhibits a trade-off between speed and accuracy. While the self-repair mechanism can handle common errors, complex dataset structures may still require manual intervention. The automatic dataset download process also struggles with anti-crawling mechanisms and login/email requests, and visual information is often lost during HTML content extraction, suggesting the need for methods that integrate neural optical understanding [Blecher et al. (2023)].

## 6 CONCLUSION

In this work, we introduced DataSEA, a fully automated system for comprehensive dataset processing, which integrates dataset search, evaluation, and analysis using large language models. Our

system allows users to input a dataset name and automatically retrieves, evaluates, and analyzes datasets from a wide range of domains. DataSEA demonstrates the effectiveness of leveraging LLMs to streamline the dataset processing pipeline, reducing manual effort and enabling researchers to focus on deeper data analysis. While our system shows promising results across diverse datasets, certain limitations such as handling databases and databanks and challenges in automatic downloads present opportunities for future work. Overall, DataSEA represents a significant step forward in automating the early stages of dataset preparation, offering researchers a powerful tool to accelerate data-driven discoveries.

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

# A APPENDIX

## MENU

## CONTENTS

This section provides a detailed description of the code structure for the DataSEA system, which is composed of three main parts: Search, Evaluate, and Analyze. Each part contains an integrated pipeline to automate the dataset processing workflow.

## A.1  SEARCH PIPELINE

The **Search Pipeline** is responsible for identifying and retrieving relevant datasets based on the user-provided input. It utilizes large language models (LLMs) to search dataset repositories and official websites. The key steps are as follows:

1. **Dataset Query**: The system accepts a dataset name as input and sends it to the LLM for generating search queries.
2. **Web Search**: These queries are used to search for datasets across multiple sources, including Kaggle, UCI, Zenodo, and official dataset websites.
3. **Link Retrieval**: The system collects potential dataset links, filtering and ranking them based on relevance and credibility.
4. **Search Output**: The final output is a set of ranked dataset links which are passed to the next pipeline for evaluation.

## A.2  EVALUATE PIPELINE

The **Evaluate Pipeline** processes the dataset links retrieved from the search phase to extract useful metadata and identify the most reliable sources. This pipeline consists of three parts:

1. **Metadata Extraction**: Using LLMs, the system extracts relevant metadata from the dataset links, such as dataset size, format, domain, and source information. This step utilizes a combination of preset and user-specified property names.
2. **Reference Paper Retrieval**: The system retrieves research papers that reference the dataset by querying academic databases. The papers are ranked by citation count, and their metadata is validated.
3. **Metadata Extension**: Metadata from the papers is integrated with the original dataset metadata to provide a more comprehensive evaluation. The system uses cross-references to ensure accuracy and consistency.

The output of this pipeline includes the final metadata and a list of reference papers, which are passed to the Analyze Pipeline.

## A.3  ANALYZE PIPELINE

The **Analyze Pipeline** is responsible for downloading, organizing, and analyzing the dataset. It performs several critical tasks:

1. **Dataset Download**: The system generates hypotheses for downloading the dataset based on metadata and content from the dataset website. For each hypothesis, it attempts to download the dataset and stores the results in corresponding folders.
2. **Code Generation**: After downloading, the system uses LLMs to generate Python code that can load, read, and visualize the dataset. The code is based on the metadata and sample points extracted from the dataset.
3. **Self-Repair**: If the generated code fails during execution, the system captures the error log and re-invokes the LLM to attempt self-repair. This step improves the robustness of the generated code.
4. **User-Defined Requirements**: Users can specify custom analysis requirements, and the system generates corresponding code to meet these needs. The code is automatically tested, and any failures are corrected using the self-repair mechanism.

The final output of the Analyze Pipeline includes a fully downloaded dataset, analysis code, and visualized sample data points. Users can interact with the system to modify or delete failed downloads as needed.

## A.4 FLOWCHART VISUALIZATIONS

The following figures illustrate the flow of the DataSEA system, both at a high level and within each individual module:

- **Figure 5**: A flowchart of the entire pipeline, from dataset input to analysis completion.
- **Figure 6**: A detailed flowchart of the Search Pipeline, showing the steps involved in dataset query and retrieval.
- **Figure 7**: A flowchart of the Evaluate Pipeline, illustrating the metadata extraction, reference paper retrieval, and metadata extension processes.
- **Figure 8**: A flowchart of the Analyze Pipeline, depicting the dataset download, code generation, and self-repair mechanisms.

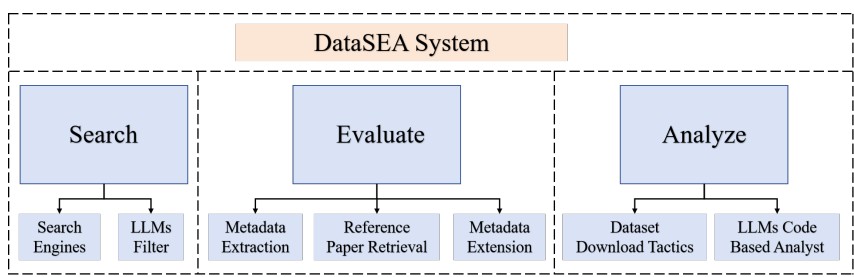

Figure 5: Flowchart of the entire DataSEA pipeline.

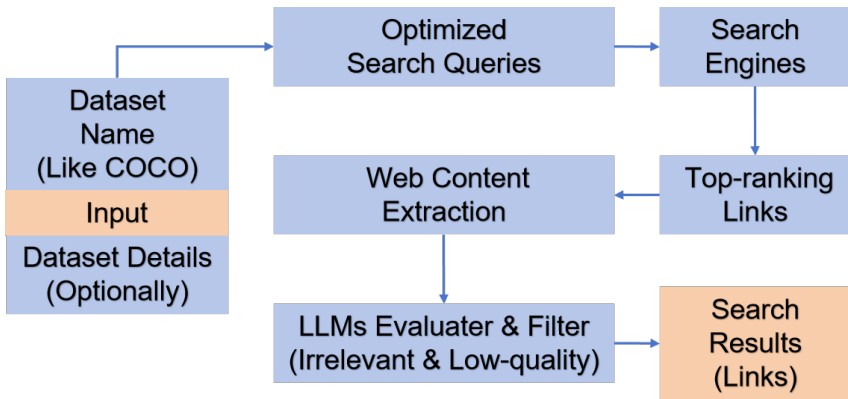

Figure 6: Flowchart of the Search Pipeline.

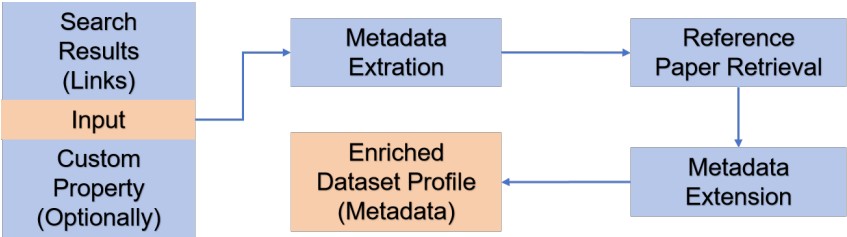

Figure 7: Flowchart of the Evaluate Pipeline.

## A.5 PROMPTS WE USE

We make tons of LLM api calls during the SEA process, and for every specific task we have a independent prompt. We will list them and show their usage.

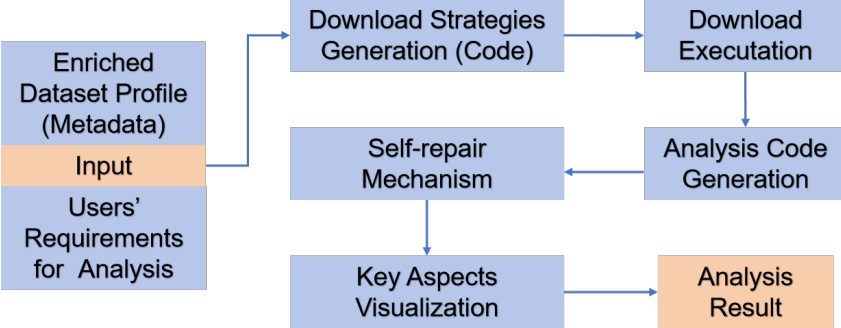

Figure 8: Flowchart of the Analyze Pipeline.

### A.5.1 DATASET WEBSITE PROMPT

The following prompt is designed to analyze a given website to determine if it contains a valid dataset download link. It is used in situations where the dataset might be located on various websites, and we need to extract the dataset link and metadata from the HTML content of the page. The prompt ensures the clarity of the task and requests the output in a structured JSON format. Below is a breakdown of the prompt:

- The prompt instructs the system to analyze the HTML of the website to determine if it is the official website for the dataset in question, identified by the dataset_name and a description.
- It asks the system to check specifically for a dataset download link and warns against mistaking an article download link for a dataset.
- If the website is identified as containing the dataset, the system should extract and return the dataset download link along with basic metadata, such as the website description.
- The prompt emphasizes returning the result in JSON format, specifying fields such as:
  - is_dataset_website: Whether the website is related to the dataset.
  - download_link_dataset: The URL link to the dataset download.
  - metadata: Any additional information extracted from the website, such as descriptions or other relevant data.
- The prompt also requests that if the dataset link is not found, the system should provide a reason for this.
- It concludes by asking for a structured JSON output without any unnecessary text or formatting, to ensure compatibility and ease of use for further analysis.

The structure of the prompt is as follows:

```
Determine whether the current website HTML is the website for the dataset "{dataset_
!!You should notice that the download link is for dataset and not article! If there
If it is, give the dataset download link from the HTML content and provide some meta

If the download link is already the dataset, then note it. Otherwise, indicate that

If it is not, provide the reason.

Return the format in JSON with the following structure:
{
    "is_dataset_website": <boolean>,
    "metadata": <object>,
    "download_link_dataset_exists": <boolean>,
    "download_link_dataset": <string>,
    "is_direct_data": <boolean>,
```

```
          "reason": <string>
      }

Note: just give the json, and do not add any extra words like adding the j-s-o-n lett

The website HTML:
"""
{html_content}
"""
```

### A.5.2 DATASET PAPER RETRIEVAL

The prompt generated by this function is used to identify whether a provided website contains the original paper for a given dataset. The original paper refers to a publication where the dataset was first introduced by the author, not merely a paper that uses the dataset. The prompt asks the LLM to carefully evaluate the HTML content of the page and determine if the paper link is present.

The model must also ensure that the link leads to a downloadable paper file (such as PDF) and not another webpage or non-relevant content. If the website does not contain the original paper, the model is required to provide a reason. The output of the task should follow a predefined JSON structure that includes flags for whether the paper link is available, and metadata about the website and the paper.

```
def generate_prompt_paper(link, dataset_name, desc = ""):
    html_content = fetch_html_from_link(link)

    if html_content is None:
        return ""

    prompt = f"""
    Determine whether the current website HTML is the website for the original paper

    Note that for "the original paper of the dataset", it means that the author of th
    It does not mean the author just use the dataset in his research, but means that

    If it is, give the paper download link from the HTML content and provide some met

    If the download link is just the paper pdf(or possibly other format) then note it

    If it is not the website of the original paper, provide the reason.

    Return the format in JSON with the following structure:
    {
        "is_dataset_paper_website": <boolean>,
        "metadata": <object>,
        "download_link_paper_exists": <boolean>,
        "download_link_paper": <string>,
        "is_direct_paper": <boolean>,
        "reason": <string>
    }

    Note: just give the json, and do not add any extra words like adding the j-s-o-n

    The website HTML:
    """
    {html_content}
    """
    return prompt
```

### A.5.3 RETRIEVING PDF LINKS FOR DATASET PAPER

The purpose of this prompt is to extract direct download links for the original dataset paper in common formats such as PDF, DOCX, or TXT from a webpage. Given the website's HTML content and dataset details, the LLM is tasked with identifying direct download links for the academic paper, filtering out irrelevant content like datasets or other material. The prompt also specifies the format for the response, which must be structured in JSON.

The returned JSON should contain download links and specify the file format, ensuring that the links are valid and directly lead to paper files rather than web pages or unrelated content.

```python
def get_potential_pdf_link(link, dataset_name, desc = ""):

    html_content = fetch_html_from_link(link)

    prompt = f"""
    I have the HTML content of a website, and I need to find any direct download lin

    The description of the paper is as follows:
    "{desc}".

    Based on this information, please search through the HTML content to find any di

    Remember you should give the direct link of paper but not other werid stuff like

    Return the format in JSON with the following structure:
    {{
        downalod_link_1: {{
            "link": "https://aaa.com",
            "format": "pdf"
        }},
        downalod_link_2: {{
            "link": "https://bbb.com",
            "format": "txt"
        }},

        ...,

        downalod_link_n: {{
            "link": "https://nnn.com",
            "format": "other format"
        }},
    }}

    Note: just give the json, and do not add any extra words like adding the j-s-o-n

    The website HTML:
    \"\"\"
    {html_content}
    \"\"\"
    """

    prompt = clamp_prompt(prompt)

    res = LLMApi(prompt)

    return res
```

### A.5.4 GENERATING DATASET METADATA EXTRACTION INSTRUCTIONS

This prompt is designed to instruct the LLM to extract relevant metadata from a collection of concatenated text files that contain information about a dataset. The LLM is provided with basic information about the dataset, such as its name and current metadata, and is tasked with extracting additional details like description, size, scale, author, and other relevant properties. The output format is strictly defined as JSON, and the LLM is asked not to provide any explanations, only the JSON data.

The key fields that the LLM is expected to populate include:

- **description**: A brief description of the dataset.

- **size**: The size of the dataset (e.g., 1GB, 10,000 samples).

- **scale**: The memory size of the dataset (e.g., 1TB, 100MB).

- **author**: The dataset's creator or author.

- **organization**: The institution responsible for the dataset.

- **usage**: Common uses for the dataset (e.g., model training, validation).

- **application fields**: Relevant application domains such as computer vision or NLP.

- **keywords**: Key terms associated with the dataset.

The LLM is instructed to use the text files as a source and output the final information in the required JSON structure.

```
# Function to create the instruction prompt without the actual text
def generate_instruction_prompt():
    dataset_name, dataset_info = read_metadata()
    prompt =f"""
    You are provided with a detailed description from a folder of concatenated text
    Your task is to extract the relevant dataset information and present it in the fo

    The basic info of dataset: its name is {dataset_name}, and its current info is {

    {{
        "dataset_name": "{dataset_name}",
        "info": {{
            "description": "<brief description of the dataset>",
            "size": "<size of the dataset (e.g., 1GB, 10,000 samples)>",
            "scale": "<scale of the memory of the dataset (e.g., 1tb, 1gb, 100mb, 10m
            "author": "<author or creator of the dataset>",
            "organization": "<organization or institution responsible for the dataset
            "usage": "<how the dataset is typically used (e.g., model training, valic
            "application_fields": [
                "<application_field (e.g., computer vision, NLP)>"
            ],
            "keywords": [
                "<keyword_1>",
                "<keyword_2>"
            ]
        }}
    }}

    Note that you should ONLY return a json file and no any other fukcing explanation

    Use the information from the concatenated text to fill out the fields as accurate
    """
    return prompt
```

### A.5.5 GENERATING DATASET REFERENCE DETECTION INSTRUCTIONS

This prompt instructs the LLM to analyze a research paper and identify whether it references a given dataset. The LLM is provided with two inputs: the name and description of the dataset, and a text string from the research paper. It is tasked with determining if the dataset is mentioned in the paper and extracting relevant details about how the dataset is used. The output is structured as a JSON object, containing information on whether the dataset is referenced and, if so, specific details on its usage and relevant text excerpts.

The key tasks for the LLM include:

- Checking if the dataset is referenced in the research paper.
- Extracting relevant information on how the dataset is used (e.g., for model training, analysis, or validation).
- Providing the specific text from the paper where the dataset is mentioned.
- Structuring the output in a JSON format, with clear fields for dataset usage, application domains, and additional details.

The prompt is designed to be comprehensive, guiding the LLM through a detailed extraction process to ensure accurate metadata is gathered from the research paper.

```
def generate_instruction_prompt(dataset_name, dataset_info):
    instruction_prompt = f"""
You are provided with two inputs:

1. A dataset named '{dataset_name}', which is described as:
   "{dataset_info}".

2. A string containing text from a research paper.

Your task is to:

- Determine if the research paper references the dataset '{dataset_name}' at any poi
- If the dataset is referenced, identify and extract the specific part of the paper
- Additionally, provide detailed information about how the dataset is used in the pap
    - Whether the dataset is used for model training, analysis, validation, compariso
    - Any specific aspects of the dataset mentioned (e.g., size, features, or unique
    - Any insights into the relevance of the dataset to the research being conducted

Your output should be a JSON object with the following structure:

{{
  "dataset_referred": <true/false>,
  "reference_details": {{
    "dataset_name": "{dataset_name}",
    "dataset_usage": "<detailed description of how the dataset is used in the resear
    "related_text": "<specific excerpt from the paper where the dataset is mentioned
    "application_field": "<application domains of the paper, in the form of a list o
    ...: any other useful info you think, can be left as blank
  }}
}}

Instructions:
- If the dataset '{dataset_name}' is not mentioned in the paper, set "dataset_referre
- If the dataset is mentioned, set "dataset_referred" to true and provide detailed in
- Ensure that "related_text" contains an exact or closely matching excerpt from the p
- If the dataset is referred to but no explicit usage is stated, provide an empty sti
"""
    return instruction_prompt
```

### A.5.6 INSTRUCTION FOR GENERATING PYTHON CODE TO VISUALIZE DATASET

This prompt is designed to instruct the language model to generate Python code for loading and visualizing a dataset. The model is guided to provide error-handling mechanisms and structured output based on dataset popularity and file availability. If the dataset is famous, libraries should be used; if not, the prompt asks the model to process local files to visualize the first 10 samples of the dataset. The prompt emphasizes proper error handling, data extraction, and visualization while logging useful information.

```python
def generate_instruction_prompt(files_info, path, error_info = ""):

    dataset_name, dataset_info = read_metadata()

    """
    Generate an instruction prompt for an LLM to generate Python code to read
    and visualize the elements of a dataset.

    Parameters:
    - dataset_name (str): The name of the dataset.
    - dataset_info (list): A list of dictionaries containing file names and the
                           head starting characters of the files (if applicable).
    """

    prompt = f"""
    # Instruction:
    Generate Python code to load the dataset '{dataset_name}', retrieve the first 10

    1. If the dataset '{dataset_name}' is famous (e.g., MNIST, CIFAR-10), use existi
    2. If the dataset is not famous, manually process the local dataset files provide
    3. Visualize the first 10 samples using matplotlib or another Python library.
    4. Ensure that all parts of the code (file loading, extraction, visualization) ha

    ## Dataset Information:
    {dataset_info}

    ## Local Dataset Files:
    {files_info}

    ### Task:
    - Write a Python program to load the dataset, extract the first 10 samples, and s
    - Write a function to visualize the samples and save the plot figure in the folde
    - Ensure all functions handle errors properly, with logs or messages.

    ### Final Output:
    You should only return plain Python code without any additional explanation!

    Ensure the code follows the below structure:
    ```python
    import os
    import matplotlib.pyplot as plt

    def load_dataset():
        try:
            ...
        except Exception as e:
            ...

    def get_first_10_samples():
        try:
```

```
            ...
        except Exception as e:
            ...

    def visualize_samples(samples):
        try:
            ...
        except Exception as e:
            ...

    def save_run_result():
        ...

    if __name__ == "__main__":
        try:
            samples = get_first_10_samples()
            visualize_samples(samples)
        except Exception as e:
            ...

        try:
            save_run_result()
        except Exception as e:
            ...
    ```

    Error log from previous code attempts: {error_info}
    """
    return prompt
```

### A.5.7 PROMPT FOR EXTRACTING DATASET DOWNLOAD LINK FROM HTML

This prompt is designed for a language model to analyze HTML content and retrieve direct or indirect download links for a dataset. The model is required to provide clear instructions on how to access the dataset, including handling any intermediate steps necessary for the download. The prompt also instructs the model to infer the file format and provide detailed instructions if the dataset cannot be directly downloaded.

```
def generate_llm_prompt(link):

    dataset_name, dataset_info = read_metadata()

    prompt = f"""
    You are tasked with analyzing the HTML content provided to identify how to downlo

    - **Dataset Name**: {dataset_name}
    - **Dataset Info**: {dataset_info}

    ### Your Objective:
    1. **Download URL**: Extract the direct download link for the dataset file if it
    2. **File Format**: Determine the file format of the dataset (e.g., zip, tar, csv
    3. **Download Steps**: Provide clear, step-by-step instructions to acquire the da
    - Clicking a direct download link.
    - Navigating to another webpage to continue the download process, if there is no
    - Completing necessary forms or accepting terms to access the dataset.
    - Any other process required to reach the final dataset.

    You should try your best to find the direct download link of the dataset. Even i
```

```
And sometimes there are direct download links but you misjudge them, so be more

NOTE!!! You should only return me a json file and do not contain any other info,

### JSON Output Format:
Present the output as a JSON object in the following structure:

```json
{{
"dataset_name": "{dataset_name}",
"download_info": {{
    "download_url": "<Direct download URL or 'None' if not available>",
    "direct_download": "<If the download url is direct or none>",
    "useful info": "<any useful infos you find, like links to potential download
    "file_format": "<File format or 'Unknown'>",
    "potential_indirect_links": "<potential download links you think>"
    "download_steps": [
    {{
        "step": 1,
        "action": "<Description of the first step needed to download the dataset>
    }},
    {{
        "step": 2,
        "action": "<Description of the second step, if applicable>"
    }},
    {{
        "step": 3,
        "action": "<Additional steps, if applicable>"
    }},
    ....,
    {{
        "step": n,
        "action": "<Additional steps, if applicable>"
    }},
    ]
}}
}}

NOTE!!! You should only return me a json file and do not contain any other info,

"""
return prompt
```

### A.5.8 PROMPT FOR GENERATING PYTHON CODE TO DOWNLOAD A DATASET

This prompt instructs the model to generate Python code that automates the process of downloading
a dataset. The model must handle both direct and indirect download links, provide error handling,
and ensure that the dataset is saved with the correct file structure. Additionally, the model is re-
quired to produce code that is generalizable and capable of managing different dataset formats and
conditions.

```
def generate_instruction(uid, idea):

    dataset_name, dataset_info = read_metadata()

    prompt = f"""
```

Write a python file to download the dataset {dataset_name}. Here are some other o

You are provided with an input dictionary stored in a variable called 'input_data
It is the info about a dataset {dataset_name}, with info {dataset_info}. Your goa
The structure of the dictionary is as follows:

{idea}

The real input is in the "input" section as this is an instruction prompt.

NOTE THAT THE dictionary is ONLY FOR REFERENCE and it may contain FALSE INFO, so

Your task is to generate Python code for the following:

- **Create a Python script file in the folder 'draft/ideas/{uid}' with the name

- **Define a function 'download_dataset()' within this file.**
    - This function should:
        - Download the dataset based on the dataset name and dataset info, and (
          - Download the dataset based on the information provided in the 'input_da
          - If you can already find infomation about the dataseat without using inp
          - Handle both direct downloads and cases where the download requires manu
          - Add try-except blocks anywhere so that the code will function normally
          - Running the download_dataset will ensure that the dataset gets downloac

    - If after trying downloading directly or indirectly(like trying all potenial
        - Print the required download steps as outlined in the 'download_steps' s
        - Output these instructions clearly so that the user can follow them to r
- **Handle direct downloads:**
    - If 'direct_download' is set to "Yes", the function should use 'requests' to

- **Create directories if necessary:**
    - Ensure that the folder 'draft/dataset/{uid}' is created if it doesn't alrea

- **Error handling:**
    - The function should check for errors during the download process, including
    - If the download fails, print a meaningful error message and proceed to try

- **Log useful information:**
    - After a successful download, print out useful metadata about the dataset fi

- **File structure and naming:**
    - Save the dataset with a filename based on the 'dataset_name' and the approp

- **Generalization:**
    - Ensure that the function is generalized to handle any properly formatted in

- **Edge cases and validation:**
    - Include validation for the existence of required fields like 'download_url
    - If a field is missing or invalid, the function should print an error and gr

NOTE THAT the download link may be a link to files like csv/txt/zip/json/... , bu

Example code structure to start:

```python
import os
import requests
```

```
def download_dataset():
    ...
    ...

    ...

if __name__ == "__main__":
    download_dataset()

Note that the example code may be wrong, so do not really rely on it. You should

You should only return python code that is content of get_dataset.py, and do not

And for the result python code, the function download_dataset, once run, will do

"""
return prompt
```

### A.5.9 PROMPT FOR GENERATING PYTHON CODE TO DOWNLOAD A DATASET

This prompt instructs the model to generate a Python script to download a dataset, with detailed instructions for error handling, logging, and alternative download methods. It ensures that the generated code is robust, handles edge cases, and can be run without any external parameters. The script also includes mechanisms to handle both direct and indirect download links, create necessary directories, and validate input data.

```
def generate_instruction(uid, idea):

    dataset_name, dataset_info = read_metadata()

    prompt = f"""

    Write a python file to download the dataset {dataset_name}. Here are some other

    You are provided with an input dictionary stored in a variable called 'input_data
    It is the info about a dataset {dataset_name}, with info {dataset_info}. Your goa
    The structure of the dictionary is as follows:

    {idea}

    The real input is in the "input" section as this is an instruction prompt.

    NOTE THAT THE dictionary is ONLY FOR REFERENCE and it may contain FALSE INFO, so

    Your task is to generate Python code for the following:

    - **Create a Python script file in the folder 'draft/ideas/{uid}' with the name

    - **Define a function 'download_dataset()' within this file.**
        - This function should:
            - Download the dataset based on the dataset name and dataset info, and (i
            - Download the dataset based on the information provided in the 'input_da
            - If you can already find infomation about the dataseat without using inp
            - Handle both direct downloads and cases where the download requires manu
            - Add try-except blocks anywhere so that the code will function normally
            - Running the download_dataset will ensure that the dataset gets downloa
```

```
        - If after trying downloading directly or indirectly(like trying all potenia
            - Print the required download steps as outlined in the `download_steps` s
            - Output these instructions clearly so that the user can follow them to r
    - **Handle direct downloads:**
        - If `direct_download` is set to "Yes", the function should use `requests` to

    - **Create directories if necessary:**
        - Ensure that the folder `draft/dataset/{uid}` is created if it doesn't alrea

    - **Error handling:**
        - The function should check for errors during the download process, including
        - If the download fails, print a meaningful error message and proceed to try

    - **Log useful information:**
        - After a successful download, print out useful metadata about the dataset fr

    - **File structure and naming:**
        - Save the dataset with a filename based on the `dataset_name` and the approp

    - **Generalization:**
        - Ensure that the function is generalized to handle any properly formatted in

    - **Edge cases and validation:**
        - Include validation for the existence of required fields like `download_url`
        - If a field is missing or invalid, the function should print an error and gi

    NOTE THAT the download link may be a link to files like csv/txt/zip/json/... , bu

    Example code structure to start:

    ```python
    import os
    import requests

    def download_dataset():
        ...
        ...

        ...

    if __name__ == "__main__":
        download_dataset()

    Note that the example code may be wrong, so do not really rely on it. You should

    You should only return python code that is content of get_dataset.py, and do not

    And for the result python code, the function download_dataset, once run, will do

    """
    return prompt
```

A.6    GOOGLE SEARCH API

We directly crawl the top-ranked links search results from google with such code:

```
import requests
```

```
from bs4 import BeautifulSoup

def search_google(query):
    # Make a request to Google Search
    url = f"https://www.google.com/search?q={query}"
    headers = {
        "User-Agent": "Mozilla/5.0 (Windows NT 10.0; Win64; x64) AppleWebKit/537.36
    }
    response = requests.get(url, headers=headers)

    # Check if the request was successful
    if response.status_code == 200:
        # Parse the HTML content
        soup = BeautifulSoup(response.text, 'html.parser')
        search_div = soup.find('div', {'id': 'search'})
        return str(search_div)
    else:
        return f"Error: {response.status_code}"

def save_to_file(content, filename):
    with open(filename, 'w', encoding='utf-8') as file:
        file.write(content)

def extract_links(html_content):
    soup = BeautifulSoup(html_content, 'html.parser')
    links = []

    # Find all 'a' tags and get their href attributes
    for a_tag in soup.find_all('a', href=True):
        links.append(a_tag['href'])

    # with open("links.txt", 'w', encoding='utf-8') as link_file:
    #         for link in links:
    #             link_file.write(link + '\n')

    return links

def get_links(input_text):
    result = search_google(input_text)
    links = extract_links(result)
    return links

if __name__ == "__main__":
    # print(get_links("scope2 dataset"))

    query = input("Enter your search query: ")
    result = search_google(query)

    if "Error" not in result:
        save_to_file(result, "search_results.html")
        print("Search results saved to 'search_results.html'.")

        # Extract links and save to a separate file
        links = extract_links(result)
        with open("draft/links.txt", 'w', encoding='utf-8') as link_file:
            for link in links:
                link_file.write(link + '\n')
```

```
        print("Links saved to 'links.txt'.")
    else:
        print(result)
```

## A.7 LONG CONTEXT INFERENCE

Long context inference involves processing large textual inputs that exceed typical token limits in language models. By employing techniques such as chunking, models can handle and analyze extensive documents without losing context or important details.

To implement long context inference, a common approach is to break down the input text into smaller chunks, process each chunk separately, and then combine the results to form a coherent output. Below is an example of Python code implementing this approach using an API to handle long texts:

```
import os
import requests
import json, sys

parent_dir = os.path.abspath(os.path.join(os.path.dirname(__file__), '..'))
sys.path.append(parent_dir)

from utils import LLMApi, clamp_prompt, clean_llm_json_res

# Get the OpenAI API key from environment variable
API_KEY = os.getenv('OPENAI_API_KEY')

# Function to split the input into chunks based on token limit
def split_into_chunks(text, max_char_len = 8888):
    chunks = []

    # Split the text into chunks of the given max_char_len
    for i in range(0, len(text), max_char_len):
        chunks.append(text[i:i + max_char_len])

    return chunks

# Function to process text of any length with chunking
def call_llm_with_chunks(instruction, text, max_tokens_per_chunk=8888, max_chunk_numk
    chunks = split_into_chunks(text, max_tokens_per_chunk)

    full_response = []

    for i, chunk in enumerate(chunks):
        if i > max_chunk_number:
            break
        print(f"Processing chunk {i+1}/{len(chunks)}...")
        prompt = generate_chunk_prompt(instruction, chunk, i)
        response = LLMApi(prompt, model=model)
        if response:
            full_response.append(response)

    return full_response

def generate_chunk_prompt(instruction, chunk, number):
    prompt = f"""
    Task: You are required to perform the following action on the provided text.
```

```
Instruction:
{instruction}

Context:
The text provided below is a portion(portion number: {number}) of a larger docume

Important Notes:
- Pay close attention to the instruction and ensure that the output reflects exac
- If the instruction requires summarizing, ensure the result is concise while ret
- If the instruction asks for rewriting, rephrase without altering the original m

Below is the text chunk that you should work on:

[Start of Text Chunk]
{chunk}
[End of Text Chunk]

Please follow the instruction precisely and produce the corresponding output.
"""
return prompt

def generate_combination_prompt(instruction, chunk_responses):
prompt = f"""
Task: You are required to combine multiple responses generated from different chu
The individual chunk responses may contain overlapping information, separate idea
Your task is to combine these responses into a single cohesive and comprehensive

The responses are results of such task: {instruction}, so merge them based on the

Below are the responses generated from different chunks. Please combine them into

"""

for i, response in enumerate(chunk_responses):
prompt += f"[Response {i+1}]\n{response}\n\n"

prompt += "Please combine the above responses into a single cohesive output, fol

return prompt

def LLM_long_api(instruction, input_text, max_chunk = 100, model="gpt-4o-mini"):
res = call_llm_with_chunks(instruction, input_text, max_chunk_number = max_chunk,
cb_pp = generate_combination_prompt(instruction, res)

return clean_llm_json_res( LLMApi(cb_pp))

if __name__ == "__main__":
res = LLM_long_api("you need to give me a story with some input info", "the story
print(res)
```

The function calls utils, and the code of util is below:

```
import json, os, requests

def change_dataset_name(name):
json_file_path = "draft/metadata.json"

with open(json_file_path, 'r') as file:
```

```
1512            data = json.load(file)
1513
1514        # Update the dataset_name
1515        data['dataset_name'] = name
1516
1517        # Save the updated JSON back to the file
1518        with open(json_file_path, 'w') as file:
1519            json.dump(data, file, indent=4)
1520
1521    def read_dataset_name():
1522        with open("draft/metadata.json", 'r') as file:
1523            data2 = json.load(file)
1524
1525        # Extract the "dataset_name" property
1526        dataset_name = data2['dataset_name']
1527
1528        return dataset_name
1529
1530    def LLMApi(input_text, max_length=8888, model="gpt-4o-mini"):
1531        api_key = os.getenv('OPENAI_API_KEY')   # Get the API key from environment variabl
1532        if not api_key:
1533            return "API key not found in environment variables."
1534
1535        url = "https://api.openai.com/v1/chat/completions"
1536
1537        headers = {
1538            "Authorization": f"Bearer {api_key}",
1539            "Content-Type": "application/json"
1540        }
1541
1542        # Clamp input text to max_length
1543        if len(input_text) > max_length:
1544            input_text = input_text[:max_length]   # Truncate the text if it's too long
1545
1546        data = {
1547            "model": model,   # Ensure you're using a valid model, e.g., "gpt-4"
1548            "messages": [
1549                {"role": "system", "content": "You are a helpful assistant."},
1550                {"role": "user", "content": input_text}
1551            ]
1552        }
1553
1554        try:
1555            # Send POST request to OpenAI API
1556            response = requests.post(url, headers=headers, data=json.dumps(data))
1557
1558            # If the response is successful (status code 200)
1559            if response.status_code == 200:
1560                result = response.json()
1561                return result['choices'][0]['message']['content'].strip()
1562            else:
1563                return f"Error: {response.status_code} - {response.text}"
1564
1565        except Exception as e:
1566            return f"An error occurred: {e}"

    def fetch_html_from_link(link):
        """Fetches HTML content from a given link."""
```

```
1566       try:
1567           response = requests.get(link)
1568           response.raise_for_status()  # Raise an error for bad responses
1569
1570           return response.text
1571       except requests.RequestException:
1572           return None  # Return None on error
1573
1574
1575   from bs4 import BeautifulSoup
1576   import requests
1577
1578   def fetch_html_from_link_no_script(link):
1579       """Fetches HTML content from a given link."""
1580       try:
1581           response = requests.get(link)
1582           response.raise_for_status()  # Raise an error for bad responses
1583
1584           html_content = response.text
1585
1586           # Try removing <script> tags from the HTML
1587           try:
1588               soup = BeautifulSoup(html_content, 'html.parser')
1589               for script in soup.find_all('script'):
1590                   script.decompose()  # Remove the <script> tags
1591               return str(soup)
1592           except Exception:
1593               return html_content  # In case of error, return the raw HTML content
1594
1595       except requests.RequestException:
1596           return None  # Return None on error
1597
1598   def clamp_prompt(long_string, char_limit=8888):
1599       if len(long_string) > char_limit:
1600           return long_string[:char_limit] + '...'
1601       return long_string
1602
1603   def read_metadata(file_path='draft/metadata.json'):
1604       with open(file_path, 'r', encoding='utf-8') as file:
1605           # Load the JSON data from the file
1606           metadata = json.load(file)
1607
1608       # Extract dataset_name and convert the entire 'info' dictionary to a string
1609       dataset_name = metadata['dataset_name']
1610       dataset_info = json.dumps(metadata['info'])  # Convert the 'info' dictionary to a
1611
1612       return dataset_name, dataset_info
1613
1614   def read_metadata_dataset_websites(file_path='draft/metadata.json'):
1615       try:
1616           with open(file_path, 'r', encoding='utf-8') as file:
1617               # Load the JSON data from the file
1618               metadata = json.load(file)
1619
1620           # Extract dataset_name and convert the entire 'info' dictionary to a string
1621           dataset_websites = metadata["dataset_websites"]
1622
1623           return dataset_websites
1624       except Exception as e:
```

```
1620            print(f"failed to read_metadata_dataset_websites, reason is : {e}")
1621            return []
1622
1623    # # Example usage:
1624    # dataset_name, dataset_info = read_metadata()
1625    # print(f"Dataset Name: {dataset_name}")
1626    # print(f"Dataset Info: {dataset_info}")
1627
1628    def clean_llm_json_res(res):
1629        res_json = res
1630        try:
1631            if res.startswith('```json\n'):
1632                res = res[len('```json\n'):].strip('` \n')
1633            # Convert the string to JSON format
1634            res_json = json.loads(res)
1635
1636        except Exception as e:
1637            # Skip invalid JSON strings
1638            print(f"Error decoding JSON for item: {res} - {e}")
1639
1640        return res_json
1641
1642
1643    def get_py_files_length(folder_path):
1644        total_length = 0
1645        # Traverse through all files in the folder and its subfolders
1646        for root, dirs, files in os.walk(folder_path):
1647            for file in files:
1648                if file.endswith(".py"):  # Only consider .py files
1649                    file_path = os.path.join(root, file)
1650                    with open(file_path, 'r', encoding='utf-8') as f:
1651                        total_length += len(f.readlines())  # Add number of lines in the
1652        return total_length
1653
1654    if __name__ == "__main__":
1655        folder_path = os.path.dirname(os.path.realpath(__file__))  # Get the current fol
1656        total_lines = get_py_files_length(folder_path)
1657        print(f"The total number of lines in all .py files (including this script) is: {t
```

## A.8  REPRODUCIBILITY

The code for the DataSEA system is available on GitHub at `https://github.com/SingleView11/DataSEA`. Detailed instructions for setting up the environment and running the pipelines are provided in the repository.

## A.9  CODE STRUCTURE

The code for the DataSEA system is organized into three main modules: Search (S), Evaluate (E), and Analyze (A). Each module contains several Python scripts responsible for different tasks within the pipeline. Below is a detailed breakdown of the file structure:

```
app/

 app.py                      # Main file to orchestrate the full pipeline
 utils.py                    # Utility functions used across modules

 S/                          # Search module
```

```
convert_json_format.py      # JSON format conversion
convert_json_format2.py     # Alternative JSON format conversion
GetRawResponse.py           # Fetch raw responses from search queries
get_firstpage_links.py      # Retrieve first-page search links
links_eval.py               # Evaluate and rank retrieved links
main_s.py                   # Main script for Search module
prompt_generation.py        # Generate search prompts for LLM
readme.md                   # Documentation for Search module
__init__.py                 # Init file for the Search module

E/                          # Evaluate module
    analyze_ref_pdfs.py         # Analyze reference papers in PDF format
    get_dataset_metadata.py     # Extract metadata from dataset sources
    get_paper.py                # Retrieve reference papers for the dataset
    get_pdfs.py                 # Download and parse PDFs
    get_sorted_ref_papers.py    # Sort and rank reference papers by citations
    longtext_api.py             # Handle long text input/output for LLMs
    main_e.py                   # Main script for Evaluate module
    main_es.py                  # Extended script for Evaluate module
    sortgs_update.py            # Update sorting logic for references
    __init__.py                 # Init file for the Evaluate module

A/                          # Analyze module
    analyze_dataset.py          # Generate analysis and visualizations for datasets
    get_download_method.py      # Determine download method for datasets
    main_a.py                   # Main script for Analyze module
    main_sea.py                 # Integrated script for Search, Evaluate, Analyze
    try_download_ideas.py       # Try different download ideas for dataset
    zip_files_final.py          # Handle final dataset packaging
    __init__.py                 # Init file for the Analyze module
```

The structure is modular, with each module containing its own set of scripts that handle specific steps in the DataSEA workflow. The modules are integrated by the app.py file, which orchestrates the end-to-end pipeline.

And below are details of using the code.

### A.10 CODE EXPLANATION

This subsection provides detailed explanations for each Python file in the DataSEA system, covering the functionality, logic, and interactions with other modules.

#### A.10.1 QUICKSTART

For setup, install requirement.txt, and make sure the openai api key is set in your environment variable.

Then run the app.py. It will ask you to input a dataset name and some optional descriptive info, and then you only need to wait for about 5-10 minutes to get a zip file that stores the infos about the dataset!

#### A.10.2 ADVANCED RUNNING

You can also do the s,e,a pipelines separately by calling s_pipeline, e_pipeline, a_pipeline function one by one, just check the main_s, main_e, main_a functions.

#### A.10.3 SEARCH MODULE (S)

**convert_json_format.py**

- **Code Usage**:

- `process_judge_info(data)`: Processes entries to parse the `judge_info` field as JSON, if possible.
- `convert_judge_info_in_file(input_file, output_file)`: Reads JSON data from `input_file`, processes it, and saves it to `output_file`.
- `eval_pipeline()`: Runs the dataset evaluation pipeline from the `links_eval` module.

**convert_json_format2.py**

- **Code Usage**:
  - `process_judge_info(data)`: Processes and converts the `judge_info` field to a valid JSON object if possible.
  - `filter_dataset_websites(data)`: Filters entries where `is_dataset_website` in `judge_info` is `True`.
  - `filter_judge_info_in_file(input_file, output_file)`: Reads JSON from `input_file`, processes and filters the entries, and saves the filtered result to `output_file`.

**GetRawResponse.py**

- **Code Usage**:
  - `google_response(query)`: Simulates a Google search by sending a search query to Google's search engine and saves the raw HTML response to `raw_search_response.html`.

**get_firstpage_links.py**

- **Code Usage**:
  - `search_google(query)`: Sends a search query to Google, parses the HTML response, and returns the search results as HTML.
  - `save_to_file(content, filename)`: Saves the provided content (HTML or text) to a file with the specified filename.
  - `extract_links(html_content)`: Extracts all the links from the provided HTML content and returns them as a list.
  - `get_links(input_text)`: Performs a Google search for the given input text, extracts the links, and returns them as a list.

**links_eval.py**

- **Code Usage**:
  - `LLMApi(input_text)`: Sends a request to the OpenAI API using the provided input text and returns the LLM's response.
  - `test(dataset_name="", desc="", need_input=True)`: Retrieves dataset links, generates prompts, and sends them to the LLM API for evaluation, returning the results.
  - `save_array_to_json(array, file_path="draft/evals.json")`: Saves an array to a specified JSON file.
  - `eval_pipeline(dataset_name="", dataset_desc="", need_input=True)`: Runs the evaluation pipeline, gathering and saving the LLM evaluations for a given dataset.

**main_s.py**

- **Code Usage**:
  - `process_judge_info(data)`: Processes the `judge_info` field, converting it to JSON if valid.

- `convert_judge_info_in_file(input_file, output_file)`: Reads JSON from `input_file`, processes `judge_info`, and writes the result to `output_file`.
- `create_folders(base_folder="draft")`: Deletes contents in the `draft` folder and creates a folder structure for storing documents and metadata.
- `create_metadata_file(base_folder)`: Creates an empty `metadata.json` file with fields for dataset metadata.
- `s_pipeline(dataset_name="", dataset_desc="", need_input=True)`: Runs the full search pipeline, including folder creation, dataset evaluation, and processing `judge_info` into JSON.

**prompt_generation.py**

- **Code Usage**:
  - `fetch_html_from_link(link)`: Fetches the HTML content from a given link. Returns the HTML as a string or `None` if an error occurs.
  - `generate_prompt(link, dataset_name, desc="")`: Generates a prompt based on the HTML content of the link and the dataset description. The prompt is used to check if the link is a dataset website.
  - `save_prompt_to_file(link, dataset_name, filename="gen_pro.txt")`: Fetches HTML, generates a prompt, and saves it to a file.
  - `clamp_prompt(long_string, char_limit=8000)`: Clamps a string to a specified character limit (default: 8000 characters).
  - `prompts_links(dataset_name, desc="")`: Fetches dataset-related links, generates prompts, and returns them as a list of dictionaries with `link` and `prompt`.
  - `test()`: Prompts the user for a dataset name, fetches the first link, and saves a generated prompt to a file.
  - `test2()`: Prompts the user for a dataset name and a specific link, then saves a generated prompt to a file.

**__init__.py (S)**

### A.10.4 EVALUATE MODULE (E)

The Evaluate module processes and extracts metadata from the dataset links obtained from the Search module.

**analyze_ref_pdfs.py**

- **Code Usage**:
  - `extract_text_from_pdf(pdf_path)`: Extracts text from a PDF file and returns it as a string.
  - `analyze_ref_papers()`: Reads research paper links from a JSON file, extracts PDF links, downloads PDFs, and runs analysis on them with the dataset.
  - `analyze_pdfs_with_dataset(folder_path, output_file)`: Analyzes PDFs in a folder by checking for dataset references and saves the results to a JSON file.
  - `generate_instruction_prompt(dataset_name, dataset_info)`: Generates a prompt for an LLM to analyze how a research paper uses the given dataset.
  - `analyze_pdf_with_dataset(text)`: Sends the extracted text from a research paper to the LLM for analysis, checking for dataset references.

**get_dataset_metadata.py**

- **Code Usage**:

    - `extract_links_from_file(file_path)`: Extracts links from a JSON file, looking for `link` and `download_link_dataset` fields.
    - `extract_all_links2(file_path)`: Extracts links from another JSON structure, including nested fields like `download_link_paper` and metadata URLs.
    - `download_files_dataset()`: Combines all extracted links from `extract_links_from_file` and `extract_all_links2`, then processes these links for downloading.
    - `download_link_content(url)`: Downloads content from a URL if the file size is less than 10MB.
    - `save_content_to_file(content, url, content_type)`: Saves the downloaded content to a file, naming it based on the URL.
    - `process_links(all_links)`: Processes a list of links by downloading content for each and saving it to the appropriate folder.
    - `extract_text_from_file(file_path)`: Extracts text from various file types (PDF, HTML, CSV, TXT) and returns the content.
    - `process_folder(input_folder, output_folder)`: Extracts and processes text from all files in a folder and saves the cleaned text to the output folder.
    - `generate_instruction_prompt()`: Generates a prompt for LLMs to extract dataset information from concatenated text.
    - `process_folder_and_generate_prompt(folder_path)`: Concatenates text from multiple files, generates an LLM prompt, and processes the results.
    - `merge_jsons(generated_data, file_path)`: Merges generated LLM results with an existing JSON metadata file.
    - `whole_pipeline_get_metadata_and_txt_info()`: Runs the entire process—downloads dataset files, processes text, generates a prompt, and merges results with metadata.

**get_paper.py**

- **Code Usage**:

    - `prompts_links(dataset_name, desc="")`: Retrieves links for potential dataset papers, generates prompts for each link, and returns a list of links with associated prompts.
    - `generate_prompt_paper(link, dataset_name, desc="")`: Generates a prompt to determine if the given link corresponds to the original paper of the dataset.
    - `get_json_evals()`: Retrieves dataset and paper-related links, generates prompts, and evaluates them using LLM.
    - `save_json_prompts()`: Retrieves evaluations from LLM for dataset and paper links and saves them in JSON format.
    - `dataset_link_prompts(dataset_name, desc="")`: Retrieves and generates prompts for dataset-related links from the `dataset_res.json` file.
    - `getValidLinks(json_path)`: Filters valid links from a JSON file based on certain criteria like `is_dataset_website`, `download_link_dataset_exists`, and `is_direct_data`.
    - `merge_link_prompts(lipros, dataset_link_prompts_array)`: Merges two arrays of link prompts, counting the occurrences of links and adding a `number` property.
    - `get_possible_papers()`: Runs the full process to retrieve, evaluate, and convert potential paper links into a JSON file.

**get_pdfs.py**

- **Code Usage**:
    - `filter_json_data(json_file, callback=None)`: Filters and returns relevant data from a JSON file based on certain paper-related attributes (`is_dataset_paper_website`, `download_link_paper_exists`, `is_direct_paper`).
    - `extract_links_and_paper_links()`: Extracts both dataset and paper download links from filtered JSON data, evaluates them, and saves them in a separate JSON file.
    - `get_potential_pdf_link(link, dataset_name, desc="")`: Fetches the HTML content of a link and generates a prompt to find direct download links for the original paper of the dataset.
    - `save_download_links_to_json(download_links_array, file_path)`: Saves the extracted download links to a specified JSON file.
    - `get_pdf_links_from_single_link(link)`: Extracts PDF links from a given URL by generating a prompt using the dataset name and metadata.
    - `download_file(link, file_path)`: Downloads the content from a URL and saves it in a specified folder. Supports formats like PDF, TXT, and CSV.
    - `download_pdfs_from_links(links, file_path)`: Downloads PDF files from a list of links and saves them to the specified folder.
    - `download_all_pdfs()`: Runs the complete process of extracting, filtering, and downloading dataset-related PDFs from the provided links.
    - `delete_all_files_in_folder(folder_path)`: Deletes all files in a specified folder.
    - `delete_all_contents_in_folder(folder_path)`: Deletes all files and subfolders within a specified folder.

**get_sorted_ref_papers.py**

- **Code Usage**:
    - `evaluate_paper(obj)`: Placeholder function for evaluating a paper. No functionality implemented yet.
    - `get_gs_rank_res()`: Reads the dataset name and calls the `sortgs_main()` function to rank results based on the dataset name.
    - `csv_to_json(csv_file, json_file)`: Converts a CSV file to a JSON format, saving the result in the specified JSON file.
    - `get_gs_papers()`: Retrieves Google Scholar ranking results for the dataset and converts them from CSV to JSON format.

**longtext_api.py**

- **Code Usage**:
    - `split_into_chunks(text, max_char_len=8888)`: Splits a long text into smaller chunks based on a character length limit.
    - `call_llm_with_chunks(instruction, text, max_tokens_per_chunk=8888, max_chunk_number=50, model="gpt-4o-mini")`: Processes text in chunks using an LLM, based on the provided instruction and model.
    - `generate_chunk_prompt(instruction, chunk, number)`: Creates a prompt for an LLM to process a specific chunk of text based on the provided instruction.
    - `generate_combination_prompt(instruction, chunk_responses)`: Generates a prompt to combine multiple LLM responses from different chunks into a single cohesive output.

- `LLM_long_api(instruction, input_text, max_chunk=100, model="gpt-4o-mini")`: Processes a long text using an LLM by splitting it into chunks, generating responses for each, and then combining the results.

**main_e.py**

- **Code Usage**:

  - `get_final_metadata()`: Combines information from various sources like dataset websites, original papers, and reference papers into the `metadata.json` file.
  - `prune_metadata()`: Refines the metadata by pruning and enhancing fields like description, size, scale, author, and usage based on evaluation data and papers. Saves the pruned metadata to `metadata_pruned.json` and updates `metadata.json`.
  - `get_prune_metadata()`: Runs both `get_final_metadata()` and `prune_metadata()` to generate and refine the metadata.
  - `e_pipeline()`: Runs the complete pipeline for retrieving papers, downloading PDFs, processing Google Scholar papers, generating metadata, analyzing reference papers, and pruning metadata.

**main_es.py**

- **Code Usage**:

  - `se_pipeline()`: Combines two pipelines, `s_pipeline()` and `e_pipeline()`, running them sequentially to process both the "S" and "E" workflows.

**sortgs_update.py**  NOTE: This code has source `https://github.com/WittmannF/sort-google-scholar`, and I update it for convenience.

- **Code Usage**:

  - `get_command_line_args()`: Parses command-line arguments for keyword, number of results, output path, sorting criteria, language filter, and other options related to Google Scholar scraping.
  - `get_citations(content)`: Extracts the number of citations from the provided HTML content.
  - `get_year(content)`: Extracts the publication year from the provided HTML content.
  - `setup_driver()`: Sets up and returns a Selenium WebDriver instance to handle Google Scholar requests.
  - `get_author(content)`: Extracts the author information from the HTML content.
  - `get_element(driver, xpath)`: Safely retrieves an element from the webpage using an XPath expression with multiple attempts.
  - `get_content_with_selenium(url)`: Uses Selenium to retrieve the page content from a URL, handling CAPTCHA challenges when required.
  - `sortgs_main()`: Scrapes Google Scholar for papers related to a dataset, extracting metadata like citations, authors, and years. Saves the results in a CSV file and optionally plots the number of citations vs. rank.

**__init__.py (E)**

### A.10.5  ANALYZE MODULE (A)

The Analyze module is responsible for downloading, organizing, and visualizing the dataset.

**analyze_dataset.py**

- **Code Usage**:

- `delete_py_files_in_folder(folder_path)`: Recursively deletes all Python (.py) files in the specified folder.
- `delete_log_json_files_in_folder(folder_path)`: Recursively deletes all JSON log files ending in _log.json in the specified folder.
- `get_analyze_code_for_all()`: Cleans up the dataset folder and generates Python code to analyze dataset files, extracting the first 10 samples and visualizing them.
- `get_file_info_list(dataset_folder, n=500)`: Reads the first 500 characters from each file in the specified folder, returning a list of dictionaries with filenames and file content.
- `generate_code_for_analyzing(files_info, path, error_info)`: Generates Python code for analyzing dataset files based on file content, dataset metadata, and past error logs.
- `generate_instruction_prompt(files_info, path, error_info)`: Generates a prompt for an LLM to create Python code for loading, analyzing, and visualizing a dataset.
- `analyze_and_run_code()`: Generates and runs Python code to analyze all dataset files.
- `analyze_and_run_code_with_self_repair()`: Attempts to run generated Python files up to three times with self-repair functionality if an error occurs.
- `regenerate_idea(file_path, e)`: Regenerates Python code for a given file if an error occurs during execution.

**get_download_method.py**

- **Code Usage**:
  - `delete_py_files_in_folder(folder_path)`: Recursively deletes all Python (.py) files in the specified folder.
  - `delete_log_json_files_in_folder(folder_path)`: Recursively deletes all JSON log files ending in _log.json in the specified folder.
  - `get_analyze_code_for_all()`: Cleans up the dataset folder and generates Python code to analyze dataset files, extracting the first 10 samples and visualizing them.
  - `get_file_info_list(dataset_folder, n=500)`: Reads the first 500 characters from each file in the specified folder, returning a list of dictionaries with filenames and file content.
  - `generate_code_for_analyzing(files_info, path, error_info)`: Generates Python code for analyzing dataset files based on file content, dataset metadata, and past error logs.
  - `generate_instruction_prompt(files_info, path, error_info)`: Generates a prompt for an LLM to create Python code for loading, analyzing, and visualizing a dataset.
  - `analyze_and_run_code()`: Generates and runs Python code to analyze all dataset files.
  - `analyze_and_run_code_with_self_repair()`: Attempts to run generated Python files up to three times with self-repair functionality if an error occurs.
  - `regenerate_idea(file_path, e)`: Regenerates Python code for a given file if an error occurs during execution.

**main_a.py**

- `a_pipeline()`: A pipeline that automates the process of:
  - `get_download_ideas()`: Retrieves ideas for how to download datasets.
  - `try_ideas_and_run_code()`: Attempts various download methods and runs the corresponding code.

- `analyze_and_run_code()`: Analyzes the dataset and runs the generated analysis code.
- `zip_folder_with_uuid()`: Zips the dataset folder with a unique identifier.

**main_sea.py**

- `sea_pipeline()`: A combined pipeline that runs both the `S+E` and `A` pipelines:
  - `se_pipeline()`: Runs both the `S` and `E` workflows sequentially.
  - `a_pipeline()`: Runs the dataset download, analysis, and packaging pipeline.

**try_download_ideas.py**

- **Code Usage**:
  - `try_ideas()`: Sets up directories, clears previous data, and iterates over dataset download ideas, attempting to generate Python scripts to download datasets based on provided ideas.
  - `generate_instruction(uid, idea)`: Generates an instruction prompt for the LLM to create Python code for downloading the dataset, handling errors, and saving the file in a specified directory.
  - `clean_code_block(code_str)`: Cleans up LLM-generated code by removing any surrounding markdown formatting (like ```python).
  - `evaluate_idea(idea)`: Uses an LLM to generate Python code for a dataset download based on the provided idea, and saves both the code and the status of the evaluation.
  - `run_all_python_files_in_folder(folder_path)`: Recursively finds and runs all Python files in a given folder and its subfolders, handling errors and logging results.
  - `try_ideas_and_run_code()`: Combines `try_ideas()` and `run_all_python_files_in_folder()` to first attempt dataset download ideas and then run the generated Python scripts.

**zip_files_final.py**

- `zip_folder_with_uuid(folder_path="draft", use_uuid=False)`:
  - This function zips the contents of a specified folder and saves it as a '.zip' file. The zip file is named using the dataset's name, and if `use_uuid` is set to `True`, a UUID is appended to the filename.
  - The zip file is saved in the `experiment_results` folder. The function ensures this folder is created if it does not exist.
  - By default, the `draft` folder is zipped, but you can specify a different folder by passing the `folder_path` argument.

**__init__.py (A)**

### A.10.6 MAIN SYSTEM COORDINATION

**app.py**

- `sea_pipeline_without_input(dataset_name, dataset_desc)`:
  - This function executes the SEA pipeline (Search, Evaluate, Analyze) without requiring user input. It accepts a dataset name and description, passing them to the respective pipeline functions `s_pipeline`, `e_pipeline`, and `a_pipeline`.
- `batch_get_experiment_res(arr)`:
  - This function takes a list of dataset names and runs the `sea_pipeline_without_input` for each dataset in the list, automating the execution of the full pipeline for multiple datasets.

**utils.py**

- `change_dataset_name(name)`:
    - This function updates the dataset name in the `metadata.json` file.
- `read_dataset_name()`:
    - Reads the `dataset_name` from the `metadata.json` file.
- `LLMApi(input_text, max_length=8888, model="gpt-4o-mini")`:
    - Sends an API request to an LLM (GPT model) with the given input text, truncating it if it exceeds the character limit.
- `fetch_html_from_link(link)`:
    - Fetches raw HTML content from a given URL.
- `fetch_html_from_link_no_script(link)`:
    - Fetches HTML content from a given URL, removing any `<script>` tags from the content.
- `clamp_prompt(long_string, char_limit=8888)`:
    - Truncates a string if it exceeds a specified character limit.
- `read_metadata(file_path='draft/metadata.json')`:
    - Reads metadata from the specified `metadata.json` file and returns the dataset name and the dataset info as a string.
- `read_metadata_dataset_websites(file_path='draft/metadata.json')`:
    - Reads the `dataset_websites` field from the metadata file.
- `clean_llm_json_res(res)`:
    - Cleans and decodes the JSON response from an LLM, removing code block formatting.
- `get_py_files_length(folder_path)`:
    - Calculates the total number of lines in all Python files in the specified folder and its subfolders.

