# OpenReview forum: "DATASEA - AN AUTOMATIC FRAMEWORK FOR COMPREHENSIVE DATASET PROCESSING USING LARGE LANGUAGE MODELS"
_ICLR.cc/2025/Conference — ICLR 2025 Conference Withdrawn Submission_

### Official Review · Reviewer_t68B · 2024-10-20

**Soundness:** 1
**Presentation:** 2
**Contribution:** 1
**Rating:** 1
**Confidence:** 5

**Summary:**

This paper introduces DataSEA (Search, Evaluate, Analyze), a fully automated framework for dataset processing using large language models (LLMs). Its goal is to streamline the data handling pipeline, reducing the manual effort involved in dataset discovery, preparation, and analysis.

**Strengths:**

This paper tries to automate the entire dataset handling pipeline, from discovery to evaluation and analysis, significantly reducing manual labor.

**Weaknesses:**

Key Weaknesses:

W1. Lack of Formal Problem Definition:

The scope of "dataset discovery" and "dataset processing" is not clearly defined in the paper. While the authors claim that "DataSEA is among the first to provide a fully integrated solution for dataset discovery, evaluation, and custom analysis using large language models," it remains unclear what specific types of datasets are targeted and how the boundaries of these processes are drawn. A formal definition of the problem and its components is essential to frame the solution appropriately.

What types of datasets does DataSEA aim to handle?
What are the specific steps involved in dataset discovery, evaluation, and analysis?
What are the inputs and outputs at each stage of the pipeline?

For example, whether the user can search the dataset like "COVID-19 cases in CA in the last season of 2022?" or "Give me datasets about house price is LA in 2023."

W2. Simplified Approach to Dataset Discovery:

The dataset discovery aspect could be seen as limited, given that existing tools like ChatGPT for Google Search or Google Dataset Search (and many others, see (https://anaconda.cloud/useful-sites-finding-datasets)) could be easily integrated to achieve similar outcomes. The paper does not explore how DataSEA improves upon or distinguishes itself from these existing dataset discovery engines, raising questions about its novelty in this area. Therefore, it is suggested to:

- Provide a comparison table showing the capabilities of DataSEA versus existing tools like Google Dataset Search and ChatGPT.
- Explain any unique features or improvements DataSEA offers over these existing solutions.
- Discuss why integrating existing tools was not chosen as an approach, if applicable.

W3. Ambiguity in Types of Evaluation:

The different types of evaluations performed by DataSEA are not well-defined. While the paper suggests that GPT can handle these evaluations, it lacks a formal breakdown or categorization of the evaluation types, leaving the reader unclear about the specific contributions and effectiveness of the system.

Can I ask questions like "how many data errors exist in the searched dataset."

W4. Undefined Scope of Custom Analysis:

The notion of "custom analysis" remains vague and insufficiently described. It is unclear what kind of analyses the system can perform and how flexible or adaptable these are to various research scenarios. Without clear examples or a formal definition, the custom analysis capability is hard to evaluate or differentiate from standard LLM tasks.

Can I ask questions like "what will the stock price trend next month?"

W5. Lack of Clear Comparisons with Baseline Methods:

Although there are many LLM-powered methods available for automated evaluation and analysis (e.g., using prompts with GPT models), the paper does not provide a comparison with state-of-the-art baselines. Without experimental results that benchmark DataSEA against these existing solutions, it is difficult to assess its effectiveness or performance improvements. The authors could compare with

- Standard LLM prompting approaches for dataset analysis,
- Existing automated data analysis tools, or
- Manual expert analysis as a human baseline

Experimentation Weaknesses:

W6. Missing Comparisons with LLM-Based Methods:

The experiments do not compare DataSEA’s performance with existing LLM-powered methods that could achieve similar results through effective prompting. This omission weakens the claims of novelty and superiority, as there is no evidence of how DataSEA outperforms such methods.

It is acceptable to include supplementary materials in the appendix. However, the formal problem definition and the proposed solutions must be clearly presented in the main body of the paper, rather than relying on simple descriptions

**Questions:**

Q1. What is the formal problem definition? Does the proposed solution address all aspects of dataset search, or are there limitations in scope?

Q2. What are the specific scopes of data evaluation and analysis in the system? Does the framework aim to address all types of data evaluation and analysis problems, or are there certain boundaries or limitations?

---

### Official Review · Reviewer_pSWy · 2024-10-31

**Soundness:** 2
**Presentation:** 2
**Contribution:** 2
**Rating:** 3
**Confidence:** 3

**Summary:**

This paper introduces a holistic and broad system, DataSEA, for dataset processing, including searching, evaluation, and analysis. This system is built on top of a set of LLMs driven by curated prompts. The author's team has conducted a series of experiments to evaluate the performance in terms of different processing speeds, and the result turned out to be promising. Code implementations are publicly available.

**Strengths:**

S1: The DataSEA system cooperates with multiple LLMs responsible for three core modules. Prompt engineering and a multi-chunk strategy are widely applied.
S2: The paper claims to be the first to provide a fully integrated solution for dataset discovery, evaluation, and custom analysis using LLMs. By automating these processes, DataSEA has the potential to significantly impact data-driven research by reducing the time spent on preliminary data handling tasks.
S3: This paper presents its objectives, the problem it aims to solve, and the solutions it proposes in a clear manner. Meanwhile, the author team has designed a bunch of experiments to support the system's robustness.

**Weaknesses:**

- Grand but Sketchy Framework: The paper presents DataSEA as a comprehensive solution for dataset preprocessing, which is an ambitious goal. However, the description of the search, evaluation, and analysis modules, while innovative, may lack in-depth observation. It's not quite convincing to me whether the system is actually useful in wild dataset processing. For instance, line 330 quotes that it takes about 3-5 mins for high-speed mode. Also, the whole system employes LLM heavily in every stages as well as integration of Google product APIs, will it leads to a high cost?

- Missing baseline: Comparative analysis is crucial for establishing the advancement of the system over existing ones. The paper could be improved by providing a section (or figure) that directly compares DataSEA's outcomes with those of other tools or frameworks mentioned in the related work section. If there are no direct competitors, the authors should explain why this is the case and how DataSEA's approach is fundamentally different or more effective.

- Poor Presentation: The figures look conceptual but lack illustration with any examples. To strengthen this part of the paper, the authors could Include case studies or use cases that demonstrate the system's capabilities in real-world scenarios. In particular to content writing, the author team should amend properly in revision:
1)  Line 99 mentioned the system is powered by a set of LLMs. But model versions seem not aligning with the experiment settings in line:324.
2) Line 158-163, the summary part appears to be suitable for the ending part of the introduction section, not the related work.
3) Improve all in-line references and labeling of every figure or table.
4) Wrong Table-of-Content in Appendix section.

**Questions:**

1. What is the model size used for Llama 3 in line 325, i.e., 8B, 70B?
2. Can you reveal more experiment details based on the domain area of datasets? The author's team discussed the limitations of the system on biological fields so far.
3. Can you describe a running example of how this system works? As the reviwer can't infer much detail from the current paper. Search: what would the optional details be? Extract: what would the custom property be? Analyze: user requirements?

---

### Official Review · Reviewer_Jtdi · 2024-11-03

**Soundness:** 2
**Presentation:** 3
**Contribution:** 2
**Rating:** 3
**Confidence:** 3

**Summary:**

The paper presents DataSEA, an automated framework for dataset processing that leverages LLMs to streamline dataset discovery, evaluation, and analysis. DataSEA consists of three core modules—Search, Evaluate, and Analyze—that autonomously handle tasks from locating datasets on the web to analyzing and generating code for custom visualization. The system aims to significantly reduce the manual labor associated with data preparation, allowing users to input a dataset name and receive automated support for finding, organizing, and analyzing data.

**Strengths:**

DataSEA  offers an agentic framework to test how LLMs can help the data processing pipeline.

The paper is easy to follow.

**Weaknesses:**

- **Real-world Relevance of the Framework**: The paper suggests that LLM agents are capable of downloading datasets, finding metadata, and locating relevant websites independently. However, major dataset platforms (e.g., Hugging Face, Kaggle) now offer easy-to-use APIs for downloading datasets and metadata. This raises the question: why test if an LLM agent can handle these tasks from scratch, rather than prompting it with relevant API code? It would be beneficial for the authors to discuss the practical use cases for DataSEA given existing tools.
- **Lack of Benchmark Focus**: The benchmark includes three components, each targeting different tasks. The search module retrieves dataset-related information, the evaluation module retrieves related literature and generates metadata, and the analysis module handles dataset downloading and visualization. However, it’s unclear what the unique challenges are for each module, which task presents the most significant bottleneck, and where the main challenges lie.
- **Dataset Size**: The evaluation dataset is relatively small, with only 100 datasets tested.
- The prompts in the appendix are poorly formatted and truncated due to page limits.

**Questions:**

See weakness.

---

### Official Review · Reviewer_KFMs · 2024-11-03

**Soundness:** 2
**Presentation:** 2
**Contribution:** 1
**Rating:** 3
**Confidence:** 5

**Summary:**

This paper focuses on data management using current techniques of LLMs, specifically, it uses first LLMs to provide a fully integrated solution for dataset discovery, evaluation, and custom analysis using large language models. It introduced the trilogy of automated dataset processing in the paper, Search, Evaluation (by LLMs), and Analyze (also by LLMs), which is an endorsable innovation in processing web data based on LLM.

**Strengths:**

This work wrapped previous work in automated dataset discovery and analysis using the power of LLMs. DataSEA here is a fully automated framework for many data engineering applications, which is a creative combination of existing tools and engineering results. On the other hand, this work is a good start toward many automated information management systems.

**Weaknesses:**

1. I'm concerned about the feasibility of this work in real applications, it seems the data processing itself is only somewhat related to tasks like document management with web search discovery, but lacks enough contribution. A comparative analysis with current or previous relevant methods could highlight DataSEA's strengths for other researchers to improve it and provide clearer insights into its performance and feasibility.

2. On the other hand, the type of data this work can support is also very limited. As pointed out in their work, it is unable to process databases, making the work have less contribution in real applications, since nearly all the existing proven-valuable data is stored in different purpose-designed databases.

3. This work heavily leaned on existing tools such as search engines and LLMs, which limits the contribution and novelty of the work. From my perspective, this paper uses LLMs and Google Search as a web crawler application combined with a few HTML tricks. It neither discussed the reliability of this framework using two existing tools nor evaluated it. In this paper, authors employ the LLMs and let them do their prompt-given jobs, if so,  might not be a good research paper, but a good technical engineering application. It would be good if authors could showcase a few pieces of innovation beyond just combining existing tools, another good point would be elaborating your novel contribution at the beginning of the paper.

4. The scope of this work seems too large and underestimates the complexity of actual web data processing. Real such data engineering means getting hands dirty from all-sourced data to efficiently store data, I'm conservative about whether LLMs are able to handle various sources, types, sizes, structures, and containers of data. I'm also curious about how authors process those huge datasets, e.g., datasets with 10GB+ single files, it would help to provide more detailed examples or some case studies showing how your system handles complex, real-world data processing scenarios.

**Questions:**

1. The multi-chunk strategy as the authors described in the paper, is a well-engineered strategy for building large-scale data RAG systems, however, there's not a universal way for chunking the dataset or documents, what specific method did you use to "breaking the data into manageable sections while maintaining context across chunks."

2. Since this is somehow like an end-to-end pipeline for dataset search on the web, would the authors mind sharing with me the overall latency of searching DataSEA? For example, suppose you are searching for the COCO dataset, regardless of the dataset download overhead (since it is related to your network bandwidth). What is the estimated waiting time I'm going to expect?

3. One particular thing I'm also interested in is what kind of specific challenge did authors faced in this work. It's never been a settled one for processing data in the wild of free forms, there must be some different tasks remaining before and after this work. The authors did not, at least I did not, fully discuss the challenges for LLMs to process data. Understanding these points would also help understand the contribution proposed in this work.

4. I'm curious about how authors process those huge datasets, e.g., datasets with 10GB+ single files, they can be binary-formatted or something else, making it extremely hard for simple general LLMs to do anything about it.

**Details Of Ethics Concerns:**

Authors seemed to give LLMs way more trust while processing data engineering on computer systems, involving unrestricted LLMs in such a field is not a good idea, I would suggest more ethical review for potential security concerns.

---

### Note · Authors · 2024-12-01

I have read and agree with the venue's withdrawal policy on behalf of myself and my co-authors.